# Stereovision-Based Ego-Motion Estimation for Combine Harvesters

**DOI:** 10.3390/s22176394

**Published:** 2022-08-25

**Authors:** Haiwen Chen, Jin Chen, Zhuohuai Guan, Yaoming Li, Kai Cheng, Zhihong Cui

**Affiliations:** 1Mechanical Engineering School, Jiangsu University, Zhenjiang 212013, China; 2Nanjing Research Institute for Agricultural Mechanization, Ministry of Agriculture, Nanjing 210014, China; 3Key Laboratory of Modern Agricultural Equipment and Technology, Jiangsu University, Zhenjiang 212013, China

**Keywords:** combine harvesters, ego-motion, FREAK feature, visual odometry, stereo camera

## Abstract

Ego-motion estimation is a foundational capability for autonomous combine harvesters, supporting high-level functions such as navigation and harvesting. This paper presents a novel approach for estimating the motion of a combine harvester from a sequence of stereo images. The proposed method starts with tracking a set of 3D landmarks which are triangulated from stereo-matched features. Six Degree of Freedom (DoF) ego motion is obtained by minimizing the reprojection error of those landmarks on the current frame. Then, local bundle adjustment is performed to refine structure (i.e., landmark positions) and motion (i.e., keyframe poses) jointly in a sliding window. Both processes are encapsulated into a two-threaded architecture to achieve real-time performance. Our method utilizes a stereo camera, which enables estimation at true scale and easy startup of the system. Quantitative tests were performed on real agricultural scene data, comprising several different working paths, in terms of estimating accuracy and real-time performance. The experimental results demonstrated that our proposed perception system achieved favorable accuracy, outputting the pose at 10 Hz, which is sufficient for online ego-motion estimation for combine harvesters.

## 1. Introduction

An increase in farming productivity and efficiency is needed to meet the challenges of population growth, climate change, and labor shortages. The development of robotics and autonomous vehicles provides an attractive solution that is core to precision agriculture [1,2]. Currently, combine harvesters are manned in most cases, and the steering accuracy decreases rather dramatically when operators have worked long hours in tedious and monotonous fields to ensure the yield is collected on time. The application of an autonomous system for combine harvesters could release the operators from arduous driving tasks, reduce operator fatigue, and improve product quality and operational safety.

Ego-motion estimation is a key capability for autonomous combine harvesters, which refers to the ability to obtain the states of combine harvesters, including locations, orientations, and velocities. These measured values combined with the target path can produce lateral and heading offset, which can be used as control variables to drive actuators (steering, throttle, or brake) to achieve autonomous navigation. Extensive research has been dedicated to localization for agricultural vehicles, with different studies varying in terms of the type of sensor used in processing methods, and accuracy, real-time performance, and cost. The Real-Time Kinematic-Global Navigation Satellite System (RTK-GNSS) is widely used in agricultural vehicles due to its accessibility and centimeter-level location accuracy [3,4,5], but it is vulnerable to interference and is incapable of perceiving the environment, limiting its usage as the primary sensor of the agricultural vehicle perception system. Inertial measurement units (IMUs) are prone to drift and can be expensive in high-precision applications. Wheel odometry does not work in rough terrain (slipping and sinking). Vision sensors are lightweight, cost-effective, and have flexible hardware settings, providing abundant information. Furthermore, visual motion estimation is less sensitive to soil mechanics and has lower drift rates.

Recent progress in computer vision, in combination with the advent of new hardware platforms, has demonstrated promising successes in a wide range of applications, such as the Mars rover, mobile robots, and self-driving cars. The latest research efforts based on visual perception in agricultural vehicles have been developed, tested, and reported [6,7]. However, as for crop field operation and combine harvesters, highly similar or repetitive patterns, varying lighting conditions, and dynamic scenes pose a great challenge to inferring accurate and timely motion from visual measurement.

This paper proposes a novel approach for estimating the ego-motion of a combine harvester using an on-board stereo camera. Our method was developed based on an open-sourced algorithm and off-the-shelf libraries in computer vision, tailored for crop field operation and combine harvesters. The primary contributions of our work are summarized as follows:An accurate and robust stereo visual odometry was developed to estimate the motion of combine harvesters. We studied the problem of implementing salient feature detection, discriminative description, and reliable matching under the agricultural scenario.We exploit prior information about the harvester motion and its environment to speed up data association. Several strategies are implemented to tackle the highly similar or repetitive agriculture scenes.Systematic and extensive evaluation of our method was performed on real datasets recorded in a crop field by a stereo camera mounted on top of combine harvesters. The results demonstrate high performance in the pose estimation task.

The remainder of the paper is organized as follows: Section 2 reviews visual-based self-motion estimation applications in agriculture. Section 3 gives an overview of our method and describes the components in detail, including stereo visual odometry (front-end) and local map optimization (back-end). Section 4 presents the experimental setup, and evaluates our method on a variety of sequences in the agricultural environment, and is followed by a discussion about its capabilities and limitations in Section 5. Section 6 concludes this work and gives ideas for future work.

## 2. Related Work

Vision-based localization is a popular and broad topic that is deeply task-dependent, and refers to the process of obtaining the position and orientation of a vehicle by analyzing consecutive images from cameras. Design and commissioning are heavily dependent on the type of motion, environmental characteristics, performance requirements, and computational resources. This section examines works that are more closely related to agricultural applications.

Agricultural scenarios are richly textured and thus suitable for feature-based vision algorithms. The effectiveness and efficiency of the feature extraction have an important influence on the performance of the visual odometer or SLAM system. SIFT [8] and SURF [9] are the most well-known algorithms and have good performance in terms of robustness to rotation, scale, and noise, but they cannot meet the real-time requirement for a VO or SLAM system due to their high computational cost. Recently, many effective and efficient feature detectors and descriptors have emerged, such as OBR [10], AGAST [11], BRISK [12], and FREAK [13]. These algorithms are implemented by encoding a local intensity value comparison/order statistic into binary strings for detection and description, meaning their detection and description are fast, and they are memory efficient.

Ericson and Åstrand [14] analyzed two visual odometry methods in different camera setups (downward-facing and forward-facing) in an agricultural scenario. Both methods use a feature-based approach. The results demonstrated that a forward-facing camera with an angle of 75° achieved the least localization error. Shahzad Zamana and Lorenzo Combab [15] presented a reliable and cost-effective monocular visual odometry system for UGV navigation in agricultural applications. The system used the normalized cross-correlation methodology to process the low-resolution images (320 × 240 pixels). Field tests were carried out on several terrains, including soil, grass, concrete, asphalt, and gravel. The experiment results verified its accuracy and robustness in motion evaluation. Dawei Jiang and Liangcheng Yang [16] designed a 3D ego-motion estimation system for an agricultural vehicle (i.e., a gantry) on uneven terrains. Harris features were tracked in image sequences using a stereo camera, and field tests were conducted on a soybean field and to demonstrate its accuracy.

To alleviate the impact of inherent camera measurement noise, two methods are prevalent: filtering approaches (EKF/UKF/Particle SLAM) and batch optimization (bundle adjustment). The work of F. Auat Cheein [17] described an optimized EIF SLAM algorithm to detect olive stems as an alternative to EKF- and UKF-based SLAM, whose linear computational cost was more suitable for real-time applications. Santosh Hiremath [18] proposed a localization and navigation method based on particle filtering to account for the different uncertainties in a semi-structured agricultural environment. Hauke Strasdat’s [19] research showed that, through progress in computer processing power and algorithms, keyframe-based bundle adjustment outperformed filtering. A plethora of excellent batch nonlinear optimization methods have emerged [20,21]. A keyframe-based sliding window strategy was used to build a metric constraint around the current frame for jointly optimizing keyframes and landmarks [22], after balancing accuracy and speed.

Feature-based VO and VSLAM are the most widely used formulations [23,24,25], and are very well-suited for richly textured agricultural environments. In general, it is difficult for visual perception systems to detect loops for applications in agricultural scenarios, and most SLAM systems will degenerate into VO systems, resulting in inevitable drift, which is still considered an unresolved problem.

## 3. Materials and Method

### 3.1. Overview

The main focus of our method concerns the way a combine harvester estimates its motion using a stereo camera. Most state-of-the-art visual perception systems make use of a multithread design to run in real time without FPGA or GPU acceleration. We follow this principle by adopting a two-thread architecture. The architecture and components of the proposed method are shown in Figure 1, and consist of a front-end and a back-end. The front-end thread detects features, creates landmarks, and outputs the poses of harvesters at a specific frame rate. The back-end thread is in charge of jointly refining landmark positions and keyframe poses at a lower rate.

As a feature-based method, the choice of feature detector and descriptor is both crucial and challenging for applications in agricultural scenarios. In addition to the essential properties required for visual odometers, i.e., rotation invariance and scale invariance, we are more concerned with robustness to similar texture and illumination changes. Furthermore, the feature should be fast to detect, describe, and match to fulfill real-time performance. We experimentally evaluated ORB, AGAST, BRIEF, etc., and adopted the FAST detector combined with the FREAK descriptor in our method, which not only demonstrated high efficiency in computation and memory usage, but also exhibited discriminative power against the repetitive agriculture scenes.

We used stereo cameras to obtain depth information from only a pair of stereo images, which greatly simplified the complexity of system initialization and avoided scale drift. As a first step to system initialization, a set of sparse landmarks are generated by means of triangulating FREAK features from the first stereo frame. The first frame is set as a keyframe and its pose as the origin of the world coordinate system (all subsequent position and coordinate estimates are made with respect to this frame).

### 3.2. Front-End

The front-end thread detects features, creates landmarks, and outputs the poses of harvesters at a frame rate. The back-end thread is in charge of jointly refining landmark positions and keyframe poses at a lower rate. The front-end refers to stereo VO for estimating the 6DOF pose of a combine harvester at each new stereo frame and deciding when to insert a new keyframe. The whole trajectory is recovered by concatenating the motion of the vehicle in successive frames. Next, we detail each step of the stereo VO, including image pre-processing, feature extraction and matching, pose estimation, new keyframe decision, and landmark creation.

#### 3.2.1. Image Pre-Processing

For each input stereo image, we perform distortion removal and rectification using known parameters to improve the efficiency of stereo matching. Then, the image is converted to 8 bpp greyscale. In addition, we perform image enhancement by means of Contrast Limited Adaptive Histogram Equalization, to normalize image brightness and to acquire the same mean and variance.

#### 3.2.2. Feature Extraction and Matching

We extract about 2000 Accelerated Segment Test (FAST) [26] features on the pre-processed stereo image, and an eight-level pyramid is used to produce multiscale features for the sake of scale invariance. In order to obtain a homogeneous distribution of the features, we divide the stereo image into a two-dimensional grid (e.g., 60 × 60 pixels). Features are extracted and matched independently over each grid cell. Harris corner response function is used to rank the features in descending order, and the top ten are retained. In practice, we use an adaptive threshold to ensure a fixed number of features. This step enables the parallel operation of feature detection and description in an efficient manner using a multicore architecture. For every retained FAST feature, we calculate its orientation (Intensity Centroid) and Fast Retina Keypoint (FREAK) [27] descriptor at each level of the image pyramid correspondingly. The FREAK descriptor adopts a retinal sampling pattern that has the overlapping receptive fields, which results in more information and more discriminative power.

After successful extraction of FREAK features, the next step is to perform stereo matching, as shown in Figure 2. For each FREAK feature in the left image, we search for its correspondences in the right image along the same row as the input stereo images are rectified. In order to generate more landmarks, a large disparity search range of half of the image width resolution is implemented. We use the Hamming distance to compute the dissimilarity measures of two descriptors. Similar textures frequently cause issues with accurate matching. We apply the right–left consistency check to eliminate mismatching. Furthermore, the distance ratio test [28] is used to reject all ambiguous correspondences when
(1)$$\frac{D\left(i,p\right)}{D\left(i,q\right)}>\tau$$
where D(i,p) denotes the Hamming distance of the best match and D(i,q) is the second-best match. An appropriate threshold can be used to reject false positives (empirically, τ=0.9). Once stereo correspondences are obtained, stereo triangulation is carried out to calculate the depth of the landmark. The coordinates of landmarks $P={\left[X,Y,Z\right]}^{T}$ are obtained as follows:(2)$$X=u_{l}\frac{B}{d},\;Y=v_{l}\frac{B}{d},\;Z=f\frac{B}{u_{l}-u_{r}}=f\frac{B}{d}\;,$$
where f is the focal length and B is the baseline of the stereo camera. The inverse of the depth Z of a pixel is denoted as $d\;={\;Z}^{-1}$, which has been proved to be favorable when visual measurement errors are modeled as Guassian distributions. For subsequent tracking of the landmarks, associated descriptors (left image descriptor) are stored for subsequent tracking. It is worth noting that only map points with a depth of less than 15 m are considered reliable for dense mapping.

Next, we establish 3D–2D correspondences by means of projecting those 3D landmarks into the current frame and searching for their 2D features. A constant velocity motion model is adopted to expedite matching, under the assumption that large motions between successive frames are prohibited because of the inertia of the harvester. The search range for each landmark is constrained to a circle of radius pixels ($r_{s}$ = 8), with the centre of the circle determined by the velocity of the previous frame. The 3D–2D correspondence is established following the same strategy as the stereo matching procedure, as shown in Figure 3.

#### 3.2.3. Ego-Motion Estimation

Due to similar and repetitive textures or moving objects in an agricultural environment, the 3D-to-2D correspondences still contain outliers, i.e., mismatched points. A popular and effective strategy, RANSAC [29], is employed to reject outliers. Next, we estimate the pose of harvesters under the condition that a set of landmarks whose 3D coordinates in the world coordinates are known, and their corresponding 2D projections in the current frame are also known. To achieve high accuracy and robustness of pose estimation, a two-stage coarse-to-fine calculation procedure is introduced. The first stage is used to obtain a coarse estimation of a pose using an EPnP algorithm [30], which is a non-iterative solution with better accuracy and less computational cost. In the second stage, the coarse estimation is used to initialize a motion-only bundle adjustment to refine the vehicle pose iteratively using all 3D–2D correspondences. As shown in Figure 4, the pose of the combine harvester in the current frame (blue rectangle) is T∈SE(3), T=[Rt0T1],R∈SO(3) and t∈ℝ3 are a rotation matrix and translation vector, respectively. Motion-only BA minimizes the reprojection error (red segment) between the projections π(·) of the landmarks Pi∈ℝ3 in the current frame and the corresponding measurements Z(·)i, including monocular Zmi=[ul vl ]T∈ℝ2 and stereo Zsi=[ul vl  ur]T∈ℝ3. i∈x denotes the set of all 3D–2D correspondences. Pose refinement can be expressed as:(3){R,t}=argminR,t∑i∈xρ(∥Zi(·)−π(·)(RPi+t)∥∑2),
where ρ is the robust Huber cost function and ∑ is the covariance associated with Zi(·). We employ a pin-hole camera projection function π defined as follows:(4)πm([XYZ])=[fxXZ+cxfyYZ+cy]=[ulvl],
(5)πs([XYZ])=[fxXZ+cxfyYZ+cyfx(X−b)Z+cx]=[ulvlur],
where (fx, fy) is the focal length and (cx,cy) is the principal point obtained from calibration. We minimize Equation (2) by performing the well-known Levenberg–Marquardt algorithm in general graph optimization (g2o) [31].

#### 3.2.4. New Keyframe Decision and Landmark Creation

Consecutive frames contain redundant information, especially with the slow motion and occasional stopping of a combine harvester. We use keyframes to cope with data redundancy, which also facilitates the nonlinear optimization in the back-end thread. Furthermore, keyframes are used to expand the map as the vehicle explores new areas. A frame is selected as a keyframe when the pose of the frame is refined successfully and the number of tracked landmarks drops below 90% of a fixed level M (typically M = 200). After a new keyframe is created, the newfound FREAK feature pairs are triangulated to create new landmarks, and both of them are added to the local map for the subsequent optimization.

**Figure 4 sensors-22-06394-f004:**
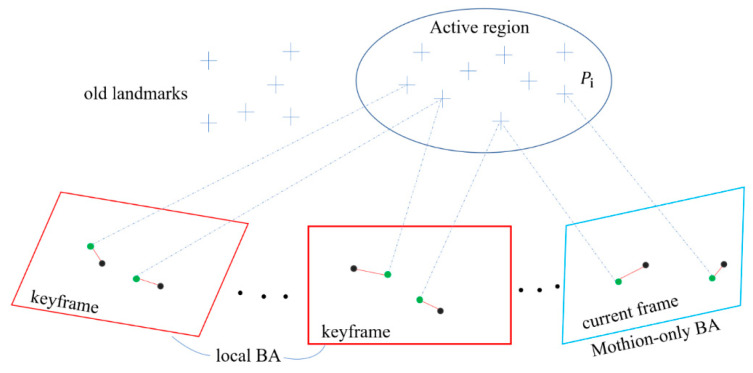
This figure depicts the process of motion-only BA and local BA. The blue rectangle denotes the current frame, the red rectangles denote the keyframe, the crosses denote a sparse set of landmarks, the green dots represent the projection of landmarks $\pi_{\left(\cdot \right)}$, the black dots represent visual measurement $Z_{i\left(\cdot \right)}$, and the red segment between green dots and black dots represents reprojection error. Motion-only BA only refines the latest vehicle pose, whereas local BA jointly optimizes keyframes and landmarks in the form of a factor graph. Both operations refer to minimizing the reprojection error.

### 3.3. Back-End

In this section, we seek to maintain and refine a local consistent map, which is an active working region around the combine harvester, consisting of a set of sparse landmarks and keyframes. When the last keyframe is inserted, the pose of keyframes and coordinates of all active landmarks are jointly optimized. This is a non-linear optimization problem tackled by least squares estimation called “Local Bundle Adjustment”, which minimizes the reprojection error between all the observed landmarks and their projection in recent neighboring keyframes. In addition to having an initial estimate relatively close to the real solution, a reasonable selection mechanism is crucial to reduce the number of parameters to be optimized. A sliding window solution is implemented to obtain $K_{f}$ active keyframes (we use $K_{f}=6$) and landmarks M visible from these keyframes, thereby achieving a constant time computational cost and real-time performance.

When a new keyframe is inserted and new landmarks triangulated from the keyframe are added, we proceed to a local bundle adjustment, which can be formulated as a Pose–Points graph, whose nodes represent the keyframe poses $\left\{K_{1},\ldots ,K_{f}\right\}$ and the landmarks $P=\left\{P_{1},\ldots ,P_{M}\right\}$, while the edges symbolise the measurement $Z_{\mathrm{ij}}$ of the $p_{i}$ landmark observed in the $K_{j}$ keyframe. Those parameters, embedded in a metric constraint space, are optimized in Equation (6) as:(6)$$\left\{P_{i},R_{j},t_{j}\right\}=\arg\underset{P_{i},R_{j},t_{j}}{\min}\sum_{i\in M}\rho \left(∥Z_{\mathrm{ij}\left(\cdot \right)}-\pi_{\left(\cdot \right)}{({\mathrm{RP}}_{i}+t)∥}_{\sum }^{2}\right),$$

Figure 4 illustrates the local BA process, as we have done in motion-only BA, where only the latest pose at the current frame is adjusted. The Huber norm is used to refine the adjacent keyframe poses and 3D positions of the landmarks in the local area. Landmarks with large reprojection errors or non-convergences are removed from the Pose–Points graph. In addition, the old keyframes and landmarks that leave the field of view of the camera are marginalized. The Pose–Points graph is updated as the vehicle moves and inserts a new keyframe.

## 4. Experimental Results

Our proposed method was tested in terms of accuracy, robustness, and real-time performance on large-scale and realistic agricultural scenarios data. In this section, details of our experimental implementation and data collection are presented, and a numerical analysis is carried out to demonstrate its capabilities and limitations.

### 4.1. Experimental Setup

The proposed method was developed for combine harvesters based on a real agricultural scene. A data acquisition platform was developed for our research, as shown in Figure 5. A wheeled combine harvester (Model: HaoLong, World Co., Ltd., Danyang, China) was employed to reproduce the trajectory, and a stereo camera (ZED, Stereolabs Inc, San Francisco, CA, USA) was mounted on top of the cabin, in the middle of the combine harvester, with a forward-looking setup at the height of 3.06 m. The image was acquired at 10 Hz with the resolution of 1280 × 720 pixels. Table 1 shows the technical specifications of the ZED. A localization unit was equipped on our platform, mounted about 10 cm behind the stereo camera on the top of the harvester, which included an RTK-GPS (Model: UM220- IV NV, PENA ELECTRON, Shenzhen, China) and an IMU (Model: MTi-10, XSENS, Enschede, The Netherlands). The accompanying location and orientation data of the GPS/IMU allowed us to compare the performance of our method. All experimental data collection was carried out by a laptop. Image data and GPS/IMU data were timestamped according to the system clock and synchronized.

The experimental dataset consisted of six stereo sequences, with a total length of 1093.8 m and a total time of around 1362 s, collected in a rice field in a farm in Danyang during the harvesting season. In order not to damage the crop and to obtain a smoother trajectory, sequences 3 to 6 were collected on stubble fields. The combine harvester was controlled manually, traversing the field in a predetermined path with an average speed of 0.8 m/s. The trajectory was recorded by the RTK-GPS/IMU sensor as the trajectory ground truth, as shown in Figure 9. Table 2 details the particularities and challenges (from easy to difficult) of our dataset.

### 4.2. Implementation Details

The proposed method was developed in C++ under Ubuntu 16.04 and ROS Kinetic, running on a standard laptop with an Intel Core i7-8550 (4C8T @ 2.60 GHz) and 16 Gb RAM. Each processing module runs over a ROS node. ROS [32] is an open-source framework that provides a flexible software architecture to allow integration of various sensors and algorithms, combined with open-sourced algorithms in computer vision and the robotics community, which led to fast and simplified development of our method. The ZED stereo camera was calibrated using the method proposed by Zhengyou Zhang [33] to obtain intrinsic and distortion parameters. Image I/O operations, feature-related operations, and EPnP were performed using the OpenCV library. Nonlinear optimization was applied using the general graph optimization (g2o) library.

### 4.3. Accuracy

The Absolute Pose Error (APE) and Relative Pose Error (RPE) [34] were used as evaluation metrics to quantify the accuracy of the estimated pose in our proposed method. Each sequence was run 10 times to alleviate the randomness of the multithreading system in the results. We used EVO (GitHub—https://github.com/MichaelGrupp/evo) (accessed on 16 June 2018) for computation and visualization. The Absolute Trajectory Errors are shown in Figure 6, and measure the overall performance of our method by computing the Euclidean distances between the estimated pose and the corresponding ground truth. Figure 7 exhibits the Relative Pose Errors of Seqs. 01 to 06, which calculate the cumulative drift over the fixed travelling distance. Our proposed method achieved favorable results on the regular harvesting paths (Seqs. 01 to 03). However, our method still encountered challenges in long-term and large-scale operations (Seqs. 04 to 06). We also compared the performance of our method with ORB-SLAM2 (GitHub—https://github.com/raulmur/ORB_SLAM2) (accessed on 22 Dec 2016), and plotted the boxplot shown as Figure 8.

Trajectories were aligned and depicted by EVO tools for both methods along with GPS ground truth, as shown in Figure 8. Our proposed method outperforms the ORB-SLAM2 on most sequences, except Seq. 06 (Figure 9). The results also show that loop closure has little effect on reducing drift, which is explained in Section 5.

### 4.4. Runtime

Achieving real-time performance is equally stringent in autonomous driving. We used Seq. 02 for timing statistics. As the self-motion estimation and local optimization are parallelly implemented in two threads, the average runtimes are given for each part separately in Table 3, given in milliseconds. By inspection of Table 3, our method outputs the pose at frame-rates of around 10 Hz, which is sufficient for online localization of combine harvesters. The front-end thread is based on detection and description computation of features with a fixed number, which has a constant computational cost per frame. In the back-end thread, the most time-consuming task is local BA, which is proportional to the number of keyframes and varies as the vehicle explores a rapidly changing scene.

## 5. Discussion

From Figure 6 and Figure 7, it can be seen that our method achieved state-of-the-art performance when the combine harvester travelled on a straight path (Seq. 01) and a curved path (Seq. 02) in a crop field. Seq. 03 contains a smooth turn, and our method obtained acceptable results. Seqs. 04, 05, and 06 contain fast and successive 90° or 180° turns, in addition to a long travelling distance, which poses a formidable challenge for accurate tracking. In these sequences, both our method and ORB-SLAM2 achieved poor results. The front-end thread of our system processes the successive stereo images to compute the incremental motion of combine harvesters, which will inevitably accumulate positional drift in long-term navigation tasks, making the estimated trajectory deviate from the actual trajectory. The back-end thread is used to refine a series of adjacent poses based on locally consistency, which is good for small or smooth scene changes. The loop closure module (place recognition) is an important component used for drift correction in SLAM [35], which is a global optimization mode. However, in the actual implementation for combine harvesters, due to the continuous change in the working environment after harvesting, along with similarity and repetitiveness of the visual appearance, it is impossible for the loop closure module to recognize places that have already been visited. Furthermore, the rotation invariance of the binary descriptor is approximated in discretized space, so the same landmarks that are observed from the opposite travel direction or a large viewing angle occasionally cannot be recognized. Further efforts are required to limit the drift.

Regarding Seq. 06, the image data were collected in a stubble field (harvested field), and contain a small amount of loop closure that can be used for ORB-SLAM2 to reduce the drift. This indicates that the result of our proposed method was not as good as that of ORB-SLAM2 in Seq. 06.

## 6. Conclusions

In this paper, we presented a system that was carefully designed to estimate the 6DOF motion of combine harvesters merely using a stereo camera, which has great advantages in terms of cost, hardware setup, and power consumption. Our method combines numerous recent advances in computer vision and robotics with a well-designed two-threaded architecture to maintain real-time performance. The key idea of our method is to continuously track and manage a sparse set of landmarks, which in turn are used to estimate the motion of the harvester. Quantitative evaluations illustrate that our method can provide an accurate and reliable estimation of the harvesters’ pose on regular harvesting paths. We expect our method will be valuable for other autonomous vehicles in agriculture.

The main drawback of our proposed approach is that pose drift accumulates over a long distance due to the absence of a loop closure module. The measurements of sequences 5 and 6 illustrate some limitations of our method. The proposed approach is a purely visual perception system, using a stereo camera as the sole external sensor. Future work will focus on fusing IMU or GPS data to eliminate drift. The robustness and accuracy of the presented method still needs to be verified in adverse weather conditions, such as wind, rain, and fog. A by-product of our method is a sparse point cloud (landmarks) map. This can be used for localization but has little use for high-level tasks such as path planning, navigation, and collision avoidance. Future work will be devoted to building a 3D dense map to represent the environment with as much detail as possible.

## Figures and Tables

**Figure 1 sensors-22-06394-f001:**
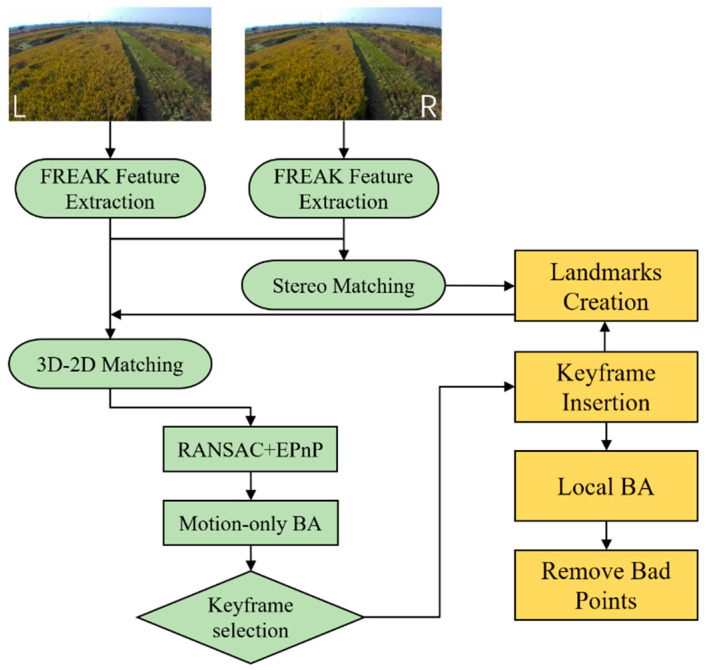
Our method is split into two separate threads, one for estimating the motion of combine harvesters (green block) and the second for optimization (yellow block).

**Figure 2 sensors-22-06394-f002:**
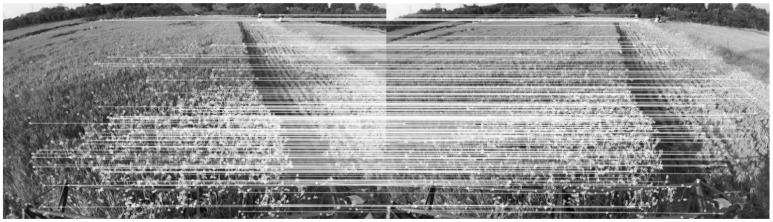
A total of 2000 FREAK features were extracted and 257 pairs were successfully matched.

**Figure 3 sensors-22-06394-f003:**
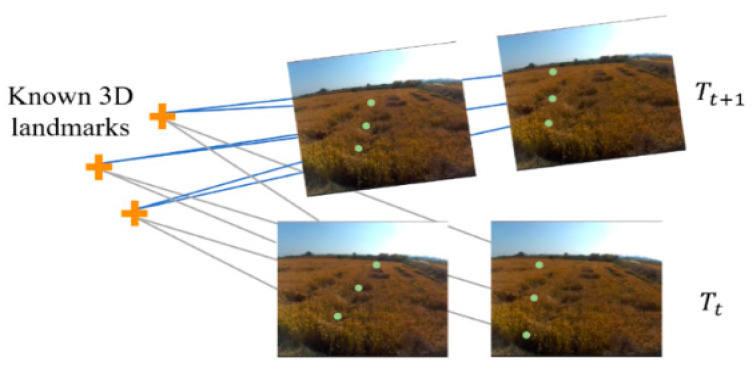
3D–2D correspondence; the crosses denote a sparse set of landmarks, the green dots represent their corresponding 2D projections in the image. $T_{t}$ and $T_{t+1}$ denote the pose at time *t* and *t*+1, respectively.

**Figure 5 sensors-22-06394-f005:**
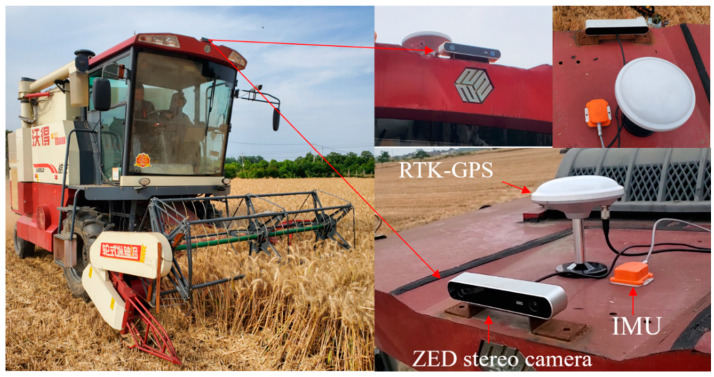
A schematic representation of the experimental platform, comprising a ZED stereo camera and RTK-GPS/IMU module as measurement instruments. A wheeled combine harvester was employed as a mobile platform.

**Figure 6 sensors-22-06394-f006:**
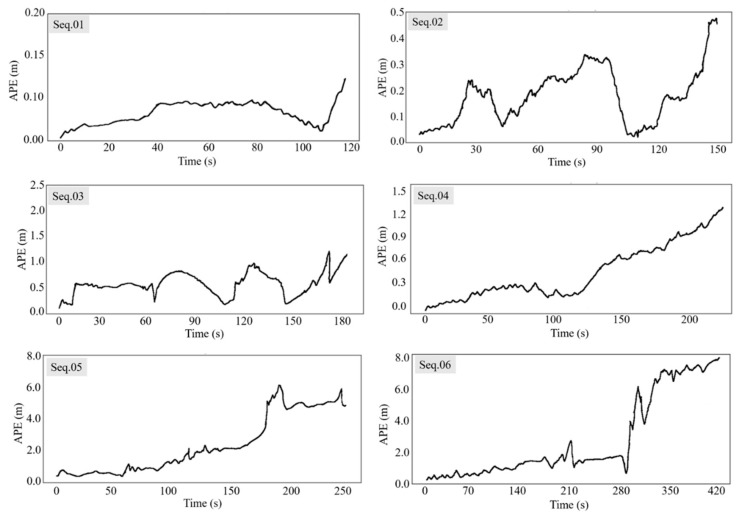
Absolute Pose Error in 6 sequences.

**Figure 7 sensors-22-06394-f007:**
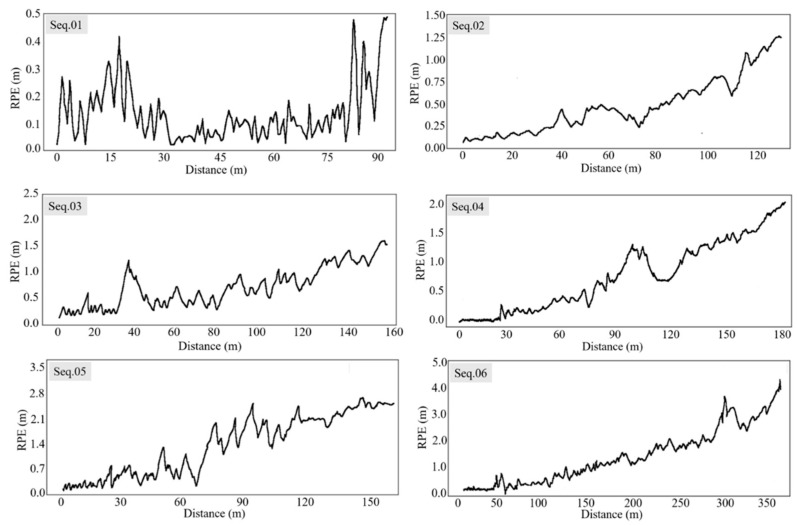
Relative Pose Error in 6 sequences.

**Figure 8 sensors-22-06394-f008:**
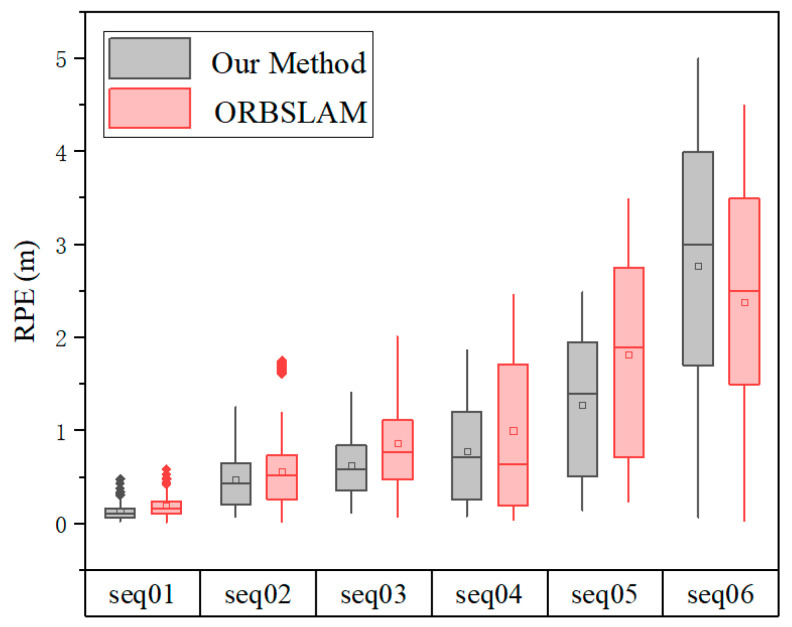
Relative Pose Error for our method and ORB-SLMA2 in 6 sequences.

**Figure 9 sensors-22-06394-f009:**
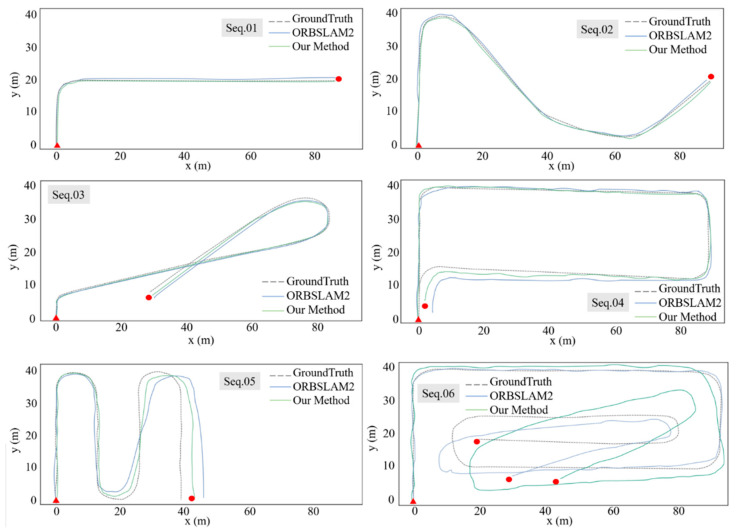
Trajectories for our method, ORB-SLAM2, and ground truth on Seqs. 01 to 06; triangles and circles highlight respectively the start and end of the trajectories.

**Table 1 sensors-22-06394-t001:** ZED technical specifications.

Image Format	2K/1080P/720P
Frame rate	60/30/15 (Hz)
Lens FOV	110°
Pixel size	2 μm
Sensor size	1/3″
Shutter	Sync. Rolling Shutter
Baseline	120 mm
Connection	USB3.0

**Table 2 sensors-22-06394-t002:** Sequence description.

Sequence #	Difficulty	Length (m)	Duration (s)	Frames	Particularities
Seq. 01	easy	96.4	116.3	3489	Straight Line
Seq. 02	easy	126.3	152.8	4584	Curved Line
Seq. 03	medium	157.9	186.3	5586	Intersecting Line
Seq. 04	medium	178.6	229.4	6882	Rectangle
Seq. 05	difficult	161.9	248.6	7458	U-shape
Seq. 06	difficult	372.7	429.4	12,882	R-shape

**Table 3 sensors-22-06394-t003:** Timing of main components of our method.

Thread	Operation	Median (ms)	Mean (ms)	Std (ms)
Front-end	Pre-processing	11.6	12.2	2.3
	FREAK Extraction	43.8	45.3	4.2
	Stereo Matching	26.5	29.1	2.8
	RANSAC	3.2	3.5	1.2
	EPnP + Motion-only BA	8.3	10.3	3.5
	Keyframe Selection	1.8	2.1	0.6
	Total	95.2	101.5	8.7
Back-end	Keyframe Insertion	10.8	10.6	2.8
	landmarks Creation	53.1	44.5	18.7
	Local BA	203.7	215.3	81.2
	Local map refinement	5.3	6.6	1.5
	Total	272.9	297.0	94.5

## Data Availability

Not applicable.

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
