# Peer review of "Stereovision-Based Ego-Motion Estimation for Combine Harvesters"

_sensors, 2022, doi:10.3390/s22176394_

Round 1
Reviewer 1 Report
Can you quantify the ambient lighting effect on the result?
Author Response
Response to Reviewer 1 Comments
Dear reviewer,
Thank you very much for giving us an opportunity to revise our manuscript, your opinions help to improve academic rigor of our paper. Based on your suggestion and request, we have studied the comments carefully and made corrections which we hope meet with approval. The main corrections are in the manuscript and the responds to the reviewers’ comments are as follows.
Point 1: Can you quantify the ambient lighting effect on the result?
Response 1: As a feature-based visual odometry, the performance of our method highly depends on the feature quality. Once we have successfully extracted the feature points, all other image information is discarded. In this work, we use FAST detector and FREAK descriptor for matching, tracking and optimization, both of them are implemented based on local intensity value comparison/order statistic, instead of intensity value themselves, which make them insensitive to illumination changes and noise. The FAST feature detector has been successfully applied in outdoor scenes and have good robustness in agricultural scenes in our method. However, at present, no matter in the literature or actual evaluation, there is still a lack of quantitative evaluation approach on the change of light.
Kind regards.
Haiwen chen
E-mail: haiwenchen1873@163.com
Corresponding author : Jin Chen
E-mail address: chenjin0009@163.com

Reviewer 2 Report
This paper presents a stereo visual odometry for the ego-motion estimation of combine harvesters. The SLAM system consists of two threads, i.e. the front-end for coarse pose estimation and the back-end for refining pose estimation. The front-end uses EPnP and motion-only BA to estimate the pose and the back-end uses local keyframe-based BA to refine the pose.
Benefits:
1) The proposed system is efficient and can be used for real-time pose estimation of combine harvesters.
2) The proposed system has been evaluated for the pose estimation of combine harvesters, and the effectiveness of the proposed method is verified in experiment.
Drawbacks:
1) The proposed method lacks enough innovation, and the algorithms of this paper have already been used in some famous visual SLAM, e.g. ORB-SLAM and ORB-SLAM2.
2) This paper do not analyze why it is superior to other methods when used to combine harvesters.
3) The experiment lacks quantitative accuracy comparison with ORB-SLAM2, and it is not sufficient to prove that the proposed method is better than ORB-SLAM2.
Author Response
Response to Reviewer 2 Comments
Dear reviewer,
Thank you very much for giving us an opportunity to revise our manuscript, your opinions help to improve academic rigor of our paper. Based on your suggestion and request, we have studied the comments carefully and made corrections which we hope meet with approval. The main corrections are in the manuscript and the responds to the reviewers’ comments are as follows.
Point 1: The proposed method lacks enough innovation, and the algorithms of this paper have already been used in some famous visual SLAM, e.g. ORB-SLAM and ORB-SLAM2.
Response 1: VO and SLAM is such a hot and broad topic that it has been used in a wide variety of applications from the Mars rover, mobile robots and self-driving cars, that is deeply task-dependent, its architectural design, parameter tuning and trade-offs between precision and speed are highly relevant to a given robot/environment/performance requirement/computational resources. Existing open-source algorithms often don't work well when deployed directly on agricultural vehicles. To our best our system has been one of the first stereo VO systems specifically designed for combine harvesters, which not only achieves competitive accuracy, but also exhibits high efficiency in memory usage and computing power, this makes our method suitable for time-critical autonomous driving.
Point 2: This paper do not analyze why it is superior to other methods when used to combine harvesters.
Response 2: Our method is based on the characteristics of the agricultural environment and combine harvesters motion type. We investigate how to extract distinctive points in images captured from agricultural scenarios with highly similar texture, efficiently and effectively. The FAST detector with FREAK descriptor is used to perform stereo matching on regular grided stereo images, for tracking evenly distributed 3D points. Some trips such as “using highly discriminative descriptors”,” adaptive threshold for high quality FAST feature”,” ratio test to reject ambiguous matching” were used in our method to improve accuracy and robustness. All of these make our method superior to other methods when used to combine harvesters.
Point 3: The experiment lacks quantitative accuracy comparison with ORB-SLAM2, and it is not sufficient to prove that the proposed method is better than ORB-SLAM2.
Response 3: The quantitative accuracy comparisons with ORB-SLAM2 have added in the experimental section, as shown below.
Kind regards.
Haiwen chen
E-mail: haiwenchen1873@163.com
Corresponding author: Jin Chen
E-mail address: chenjin0009@163.com

Reviewer 3 Report
In the manuscript titled " Stereovision-Based Ego-Motion Estimation for Combine Harvesters", the authors explored an important point, overall, I think the manuscript provides sufficient information for the scientific community, and in my opinion, it can be accepted for publication after minor revision.
Following are some specific comments.
1. The authors must check the manuscript from the orthographic point of view and for English language.
2. The organization and presentation of this work need to improve.
3. The introduction part should be made more elaborative via discussion on different conventional methods.
4. The conclusion section should be precise with more information.
Author Response
Response to Reviewer 3 Comments
Dear reviewer,
Thank you very much for giving us an opportunity to revise our manuscript, your opinions help to improve academic rigor of our paper. Based on your suggestion and request, we have studied the comments carefully and made corrections which we hope meet with approval. The main corrections are in the manuscript and the responds to the reviewers’ comments are as follows.
Point 1: The authors must check the manuscript from the orthographic point of view and for English language.
Response 1: After careful examination, some spelling errors were found and corrected in the original manuscript.
Point 2: The organization and presentation of this work need to improve.
Response 2: Some sentences and expressions have been reorganized in the original manuscript.
Point 3: The introduction part should be made more elaborative via discussion on different conventional methods.
Response 3: We have added content in the introduction section. More discussion on different conventional methods is shown in Related work section.
Point 4: The conclusion section should be precise with more information.
Response 4: The quantitative accuracy comparisons with ORB-SLAM2 have added in the experimental section, as shown below.
Kind regards.
Haiwen chen
E-mail: haiwenchen1873@163.com
Corresponding author : Jin Chen
E-mail address: chenjin0009@163.com

Reviewer 4 Report
The authors have an interesting idea of using camera sensors for autonomous combine harvesters. However, the authors should address the following issues:
1. How could the system help to navigate the combine harvesters autonomously? It should be clearer in the introduction section.
2. In line 69, the authors said: “several strategies are implemented to tackle the highly similar or repetitive agriculture scenes”. However, the authors did not present these strategies clearly. Lines 145 to 147, and 221 to 223 did not show why FREAK and RANSAC are more suitable than other methods.
3. Why do the authors implement the system on Ubuntu 16.04, which is quite old? it means that the time for applying this system in the future could be very short.
4. In line 346, the authors said that the proposed method outperformed ORB-SLAM2. However, in Figure 8, the last figure shows the result of the proposed method was not as good as the result of ORB-SLAM2. The authors did not explain why it was not good.
5. It should have a paragraph explaining why the proposed method has better performance than ORB-SLAM2 in the discussion.
6. Why author did not compare the execution time between the proposed method to ORB-SLAM2 to prove its high performance?
7. Many SLAM programs are designed in multi-thread architecture: front-end, back-end, viewers, and using the same algorithms, what is your contribution to program structure? why is your program better? Which makes your program suitable for combining harvesters and the outside environment? You should compare your program structure to ORB-SLAM2 or others to make it clearer and prove your contribution.
8. In line 135 proposed mothed -> proposed method
Author Response
Response to Reviewer 4 Comments
Dear reviewer,
Thank you very much for giving us an opportunity to revise our manuscript, your opinions help to improve academic rigor of our paper. Based on your suggestion and request, we have studied the comments carefully and made corrections which we hope meet with approval. The main corrections are in the manuscript and the responds to the reviewers’ comments are as follows.
Point 1: How could the system help to navigate the combine harvesters autonomously? It should be clearer in the introduction section.
Response 1: The classical autonomous systems can be divided into perception module and action module, The perception module refers to calculate the lateral or longitudinal or heading offset of vehicles as intermediate variables to feed the action module, including proprioception (position, orientation, velocity, etc.) and exteroception (crop row, cut edge, and obstacle, etc.), which relies heavily on raw sensory inputs, such as GPS, IMU, camera, and LiDAR, etc. The action module receives lateral offset and heading angle generated by the perception module, and calculates and sends the steering angle to the actuators (steering, throttle, or brake).
For simplicity's sake, we add it in the Introduction Section: “Ego-motion estimation is a key capability for autonomous combine harvesters, which refers to the ability to obtain the states of combine harvesters, including locations, orientations, and velocities. These measured values combined with the target path can produce lateral and heading offset, which will be used as control variables to drive actuators (steering, throttle, or brake) to achieve autonomous navigation.”
Point 2: In line 69, the authors said: “several strategies are implemented to tackle the highly similar or repetitive agriculture scenes”. However, the authors did not present these strategies clearly. Lines 145 to 147, and 221 to 223 did not show why FREAK and RANSAC are more suitable than other methods.
Response 2: Several strategies mainly include “using highly discriminative descriptors”,” adaptive threshold for high quality FAST feature”,” ratio test to reject ambiguous matching”. These little tips refine the tracing process and improve the robustness of the system.
In this paper, we use FAST detector and FREAK descriptor, the most important consideration is its discrimination power, as the adopts retinal sampling pattern that has the overlapping receptive fields, which brings more information and more discriminative power. On the one hand, this can be confirmed in the literature, on the other hand, we have also done tests on images of agricultural scenes, as shown below:
RANSAC is a very effective method to remove outliers, which is commonly used in VO and SLAM systems.
Time consumption (ms)
|
Operation |
Methods |
Feature extracted per frame |
||||
|
500 |
1000 |
1500 |
2000 |
2500 |
||
|
Detector |
FAST |
5.36 |
9.27 |
13.62 |
16.94 |
20.14 |
|
AGAST |
6.14 |
9.87 |
14.34 |
17.78 |
22.02 |
|
|
AKAZE |
15.82 |
26.41 |
40.18 |
46.82 |
62.13 |
|
|
GFTT |
420.12 |
860.42 |
1152.86 |
1489.11 |
1799.18 |
|
|
Descriptor |
BRIEF |
7.92 |
14.12 |
20.98 |
26.29 |
33.19 |
|
BRISK |
15.24 |
29.17 |
43.29 |
52.79 |
65.31 |
|
|
FREAK |
8.72 |
15.92 |
23.58 |
28.36 |
34.29 |
|
Successfully matching rate (%) (*our method)
|
Extractor |
Feature extracted per frame |
||||
|
500 |
1000 |
1500 |
2000 |
2500 |
|
|
FAST+BRIEF |
40.28 |
20.18 |
13.42 |
10.24 |
8.04 |
|
ORB |
|
|
|
|
|
|
BRISK |
56.12 |
28.76 |
18.67 |
14.13 |
11.23 |
|
FAST+FREAK* |
51.48 |
26.22 |
17.13 |
12.85 |
10.40 |
Point 3:Why do the authors implement the system on Ubuntu 16.04, which is quite old? it means that the time for applying this system in the future could be very short.
Response3: When we cloned orb-slam2 from Github, it was easier to deploy on Ubuntu16.04, and our approach was also implemented on 16.04 for comparison purposes contain significant loop-closures.
Point 4:In line 346, the authors said that the proposed method outperformed ORB-SLAM2. However, in Figure 8, the last figure shows the result of the proposed method was not as good as the result of ORB-SLAM2. The authors did not explain why it was not good.
Response 4: we designed a stereo VO for combine harvester, the main consideration is that the loop closure module is hard to recognize places that have already been visited, because, usually, the harvester operates like an endless corridor, where it does not return to the origin, or when it does return to the origin, the scene has changed, in which case SLAM will degenerate into VO. But as for the Seq. 06, the image data was collected at a stubble field (harvested), it contains a small amount of loop closure can be used for ORB-SLAM2 to reduce the drift, it explains that our proposed method was not as good as the result of ORB-SLAM2 in Seq. 06.
Point 5: It should have a paragraph explaining why the proposed method has better performance than ORB-SLAM2 in the discussion.
Response 5: we have revised the sentence in discussion, it can clearly be seen that performing full SLAM has a very small effect on decreasing long-term drift, however, adding the loop closure modular comes at increased computational cost, the overall computational budget required more than doubles.
Point 6:Why author did not compare the execution time between the proposed method to ORB-SLAM2 to prove its high performance?
Response 6: When we tested ORB-SLAM2 in an agricultural scenario, based on its default Settings, it was no surprise that it was able to output the pose information at 30Hz, which is consistent with his original statement. When a VO or SLAM is deployed on agricultural vehicles with slow speed, the high frame rate will lead to redundant information and increase the computational burden without improving the accuracy. For a combine with an average forward speed of 1 m/s, a pose output of 10hz is appropriate
Point 7:Many SLAM programs are designed in multi-thread architecture: front-end, back-end, viewers, and using the same algorithms, what is your contribution to program structure? why is your program better? Which makes your program suitable for combining harvesters and the outside environment? You should compare your program structure to ORB-SLAM2 or others to make it clearer and prove your contribution.
Response 7: VO and SLAM is such a hot and broad topic that it has been used in a wide variety of applications from the Mars rover, mobile robots and self-driving cars, that is deeply task-dependent, its architectural design, parameter tuning and trade-offs between precision and speed are highly relevant to a given robot/environment/performance requirement. Our method is based on the characteristics of the agricultural environment and designed specifically for combine harvesters, outperformed ORB-SLAM2 in most cases. The quantitative comparisons have added in the experimental section, as shown below.
Point 8:In line 135 proposed mothed -> proposed method.
Response 8: This has been corrected in our manuscript.
Kind regards.
Haiwen chen
E-mail: haiwenchen1873@163.com
Corresponding author : Jin Chen
E-mail address: chenjin0009@163.com

Round 2
Reviewer 2 Report
The proposed method lacks enough innovation, and the algorithms of this paper have already been used in some famous visual SLAM, e.g. ORB-SLAM2 and ORB-SLAM3.
Reviewer 4 Report
It is better that the response to the reviewers' comments includes the line numbers in which the revised contents are.
The revised paper should reflect the responses.